# Current Situation and Future Development Direction of Soil Covering and Compacting Technology under Precision Seeding Conditions in China

**Wenchao Yang** [1,2], **Jin He** [1,2,*] , **Caiyun Lu** [1,2] , **Han Lin** [1,2] , **Hanyu Yang** [1,2] and **Hang Li** [1,2]

1 College of Engineering, China Agricultural University, Beijing 100083, China; ywc@cau.edu.cn (W.Y.); lucaiyun@cau.edu.cn (C.L.); linhanctrc@cau.edu.cn (H.L.); s20213071335@cau.edu.cn (H.Y.); hli@cau.edu.cn (H.L.)
2 Key Laboratory of Agricultural Equipment for Conservation Tillage, Ministry of Agricultural and Rural Affairs, Beijing 100083, China
* Correspondence: hejin@cau.edu.cn; Tel.: +86-010-6273-7300

**Abstract:** Precision seeding is an effective technical way to improve quality and reduce material costs. However, certain factors limit the development and promotion of precision seeding technology, among which the main factor is that the agronomic requirements of sowing vary significantly in different areas, and there is a lack of suitable implements; at the same time, each link among the parts interact, which brings specific difficulties to improving the operational effect. This paper summarizes the precision seeding technology from the perspective of precision seeding realization and the consistency of seed spacing and use of consistent sowing depth control technology. A detailed description is included of the precision seed-discharge technology, the smooth seed-dropping technology, and the seed falling into the seed bed with precision. The need for regulation technology for consistent sowing depth is explained from a developmental perspective. The research status of compatible sowing depth control technology is described, as used in soil covering and compacting. This paper presents the problems and future development directions of soil-covering and compacting technology under precision-seeding conditions, including developing adaptable regional soil-covering and compacting devices, improving the processing and equipment technology of enterprises, strengthening the structural optimization design of soil-covering and compacting devices, conducting research into soil-covering and compacting control methods, and promoting the intellectual development of agricultural equipment.

**Keywords:** precision seeding; consistent sowing depth; soil covering and compacting; developmental direction

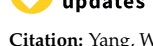



## 1. Introduction

With the constant advancement and refinement of agricultural technology in recent years, higher standards have been placed on existing sowing techniques to reduce material costs for sowing operations and increase crop yields [1]. Precision seeding, as a high-demand method of operation, has been widely employed in sowing crops such as corn and soybeans [2].

Precision seeding involves placing seeds into the seed bed using a supporting precision seeding machine to ensure precise sowing volume, uniform sowing depth, and even grain spacing, resulting in flushed, solid, and consistent seedlings, which increases yield and reduces costs. The operation of this machine has a direct impact on the quality of machine sowing, as well as the level of crop yield and quality (Figure 1). To enhance the yield potential over a large area, as well as a group of mechanized operations, the Ministry of Agriculture and Rural Affairs released a "notice on further improving the quality of mechanized sowing", which requires all agricultural and rural departments to prioritize

the work of grain and oil production and agricultural mechanization by enhancing the quality of machine sowing [3].

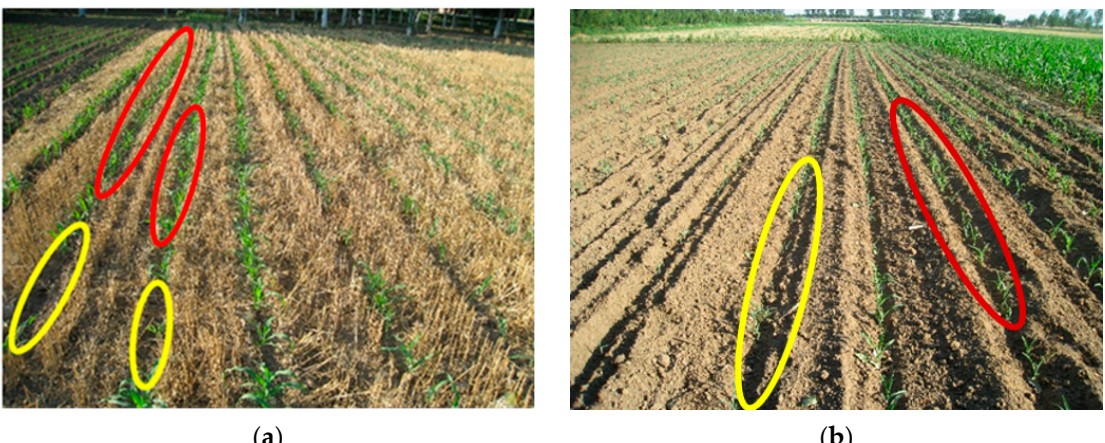

(a)

(b)

**Figure 1.** Uneven condition of the seedling buds. (**a**) The seedling buds are uneven from top to bottom. (**b**) The seedling buds are uneven from left to right.

Precision seeding primarily involves technical coordination and control of precise seed discharge, smooth seed guidance, and the seeds falling into the seed bed with precision and at a consistent sowing depth. As a critical reference index for evaluating precision seeding, constant sowing depth refers to the uniform distance of seeds from the surface of the seed bed after soil covering and compacting operations, to meet the agronomic operation sowing depth index requirements [4–6]. Therefore, to ensure sowing quality and achieve precision seeding, not only is it necessary for the seed metering device to operate stably and provide a uniform seed flow, but also for the devices that are in contact with soil to operate stably [7], the soil covering device to cover the soil consistently during the soil covering process, and the soil flow to cover the seeds evenly and effectively; furthermore, the compacting pressure of the compacting device can be adjusted, according to the soil condition and agronomic requirements [8].

Kinsner et al. demonstrated that sowing depth directly affects the emergence of wheat seedlings, which is 80% when the sowing depth is 55 mm but drops to 70% when the sowing depth is 35 mm or 80 mm. Proper and uniform sowing depth has a crucial impact on the emergence of seedlings and subsequent crop development [9]. Rivera et al. found that seeds sown too shallowly during seeding operations faced an increased risk of being ingested by birds and insects, resulting in reduced grain yield. At the same time, seeds sown too deeply made it challenging to break the seeds, and slowed the development of emergence, reducing the ability of seedling buds to compete with weeds after germination [10]. Therefore, in-depth research on the soil-touching parts that control the vertical distribution of seeds in the soil is necessary to explore more advanced means and methods to study the mechanism of soil components and soil action during precision seeding to provide accurate and reliable technical parameters and to provide better performance of soil-touching parts for the precision seeder. The devices in contact with the soil have a direct impact on the spatial arrangement of seeds in the seed bed, and the precise placement of the seeds in the seed bed is achieved through the regulation of soil covering, compacting, and other operational aspects to achieve a good yield and a good harvest of grain. The devices in contact with soil directly impact the spatial arrangement of the seeds that fall into the seed bed. By regulating the soil covering and compaction procedures and other operational aspects, precision seeding achieves accurate seed placement within the seed bed, ultimately promoting optimal seed development and enhancing crop yield potential [11].

In conclusion, the devices in contact with the soil, such as covering and compacting devices, play a crucial role in ensuring and enhancing the quality of precision seeding technology for achieving consistent sowing depth [12]. Therefore, future research efforts

should focus on improving the quality of soil covering and enabling real-time regulation of soil compacting pressure to ensure a stable operation at each stage of the process and achieve the goal of precision seeding.

## 2. Precision Seeding Technology

Precision seeding technology was first proposed in the 1940s by the first foreign researchers, while the theory of precision seeding has also been rapidly developed [13]. In the 1960s, Britain, Germany, and other countries researched precision sowing technology; CASE, KINZE, John Deere, and other large foreign companies in the relevant precision seeding technology research are perfect examples [14]. China started the technical studies and the related machinery development of precision seeders at the end of the 1980s [15]. Due to factors such as the technical level and price sales of the agricultural machinery factory at that time, the early precision seeder was mostly a semi-precision and small precision seeder.

Precision seeding mainly includes precise seed discharge technology, smooth seed-dropping technology, the seed falling into the seed bed with precision, and technology for the consistent regulation of sowing depth [16] (Table 1). At present, the research on precision seeding mainly involves precision seed-discharge technology, smooth seed-dropping technology, and the seeds falling into the seed bed with precision, which has a significant influence on the consistency of grain spacing; because of the necessity of consistent regulation of sowing depth, the two aspects of soil covering and compacting technology are essential means of controlling the thickness of sowing depth.

**Table 1.** Precision seeding: operational technology process.

| | Application Scenario | Major Technology | Each and Every Link | Influence Results |
|---|---|---|---|---|
| The precision seed-discharge technology | Seed metering device | Mechanical seed metering devices, pneumatic seed metering devices | Discharge of seeds | The consistency of seed spacing |
| The smooth seed-dropping technology | Seed-guiding device | Conveyor-belt-type seed guide, brush-belt-type seed guide | Seed path orientation | |
| The seeds falling into the seed bed with precision | Seed bed | Seed-pressing wheel, seed-pressing tongue | The fall into the seed bed | |
| The consistent sowing-depth regulation technology | The devices in contact with soil | Terrain-adapting furrow opener, soil covering and compacting | Covering seeds | The sowing-depth-consistency regulation |

### 2.1. The Seed-Spacing Consistency-Regulation Technology

As a critical component of precision seeding [17], the seed metering device has high requirements for seed planting performance to ensure that there will be no re-sowing and no leakage and that the seeds are discharged smoothly and evenly during the seeding operation. There are mainly two mechanical and pneumatic seed metering devices for precision seeding [18]. Mechanical seed metering devices have a simple structure. Nevertheless, compared to pneumatic seed metering devices and seeding accuracy in general, the requirements for seeds are high; it is easy to cause damage to the seeds, and these devices are primarily used in low-speed sowing operations. The commonly used mechanical precision seed metering device mainly includes the tilting-disc type [19–21], spoon-wheel type [22], and finger-clip type [23] (Figure 2).

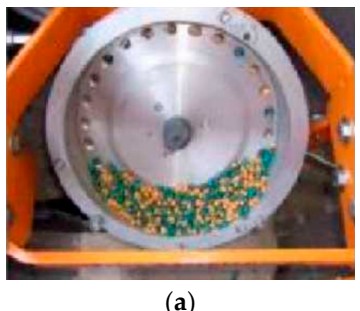
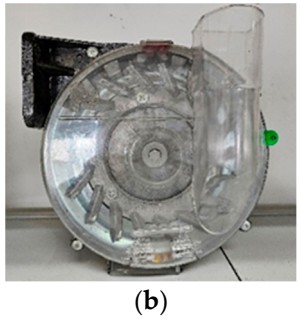
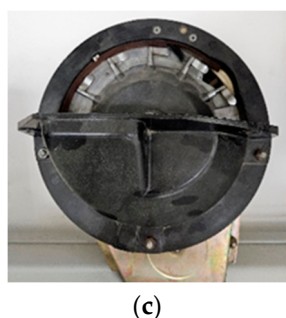

(**a**)          (**b**)          (**c**)

**Figure 2.** Mechanical seed-metering devices. (**a**) Tilting-disc type. (**b**) Spoon-wheel type. (**c**) Finger-clip type.

Among them, the structure of pneumatic seed metering devices is more complex, requiring a pneumatic system and transmission system to complete the operation. They have a high sowing accuracy and are primarily used in medium and large sowing machines; and the foreign developed model is more perfect, and mainly used in the high-speed precision seeder [24]. The commonly used pneumatic precision seed metering devices mainly include the pneumatic-suction type, the pneumatic-pressure type, the air-blown type, and the central set of row type [25–28] (Figure 3).

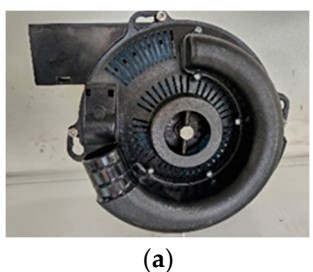
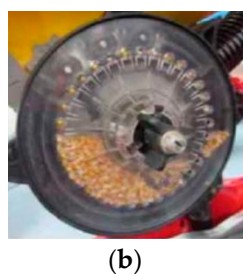
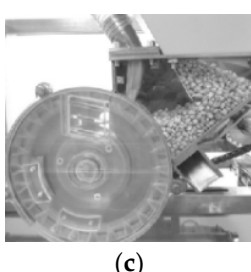
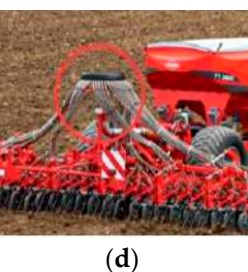

(**a**)          (**b**)          (**c**)          (**d**)

**Figure 3.** Pneumatic seed metering devices. (**a**) Pneumatic-suction type. (**b**) Pneumatic-pressure type. (**c**) Air-blown type. (**d**) Central set-row type.

For example, the 'vacuum meter' pneumatic-suction type seed metering device developed by the Kinze Company has a seed metering disc with multiple seed-absorbing nests to reduce the rotational speed of the seed metering disc and extend the seed filling time. At the same time, an electric motor drives the seed-metering disc to ensure stable rotation to improve the seed-filling and cleaning effect. The operational speed can reach 12.8 km/h while providing quality seed discharge [29]. Heilongjiang BeiKeRuiSi Modern Agricultural Technology Co (Manufacturer: Heilongjiang Backace Modern Agro-Tech Co., Ltd.

City: Nangang District, Harbin City, Heilongjiang Province.China.). researches a pneumatic-suction type seed metering device, which is driven by a motor, with a rotating speed of 0~150 r/min, an air pressure of 6~8 kPa, a seed-discharge disc and a rotating hub which is airtight and solid and has a synchronous rotation; the rotating hub and the fan produce a negative pressure through the pneumatic suction to achieve seed discharge using gas suction. It has the advantages of a compact structure and stable operation for solving problems such as the seeds falling off, poor air tightness, and varying seed-discharge-disc rotation speed, caused by mechanical vibration in a high-speed no-tillage operation [30].

Shi Song designed a pneumatic combination hole-type precision corn-seed metering device to realize the improvement in seed-filling performance of the seed metering device under high-speed operation; this improves seed mobility under positive pressure of the airflow, removes the seed-stirring mechanism of traditional machinery, has a simple structure and works stably in a high-speed operation [31]. Yang Z. G. carried out a simulation and experimental research on an air-blown corn-seed metering device, mainly conveying seeds to the seed-drop port through the typed hole on the rowing wheel. During the operation,

the airflow at the mouthpiece can blow away the excess seeds to realize single seeding [32]. Salavat Mudarisov established a mathematical model of the distribution system of the central set-row seeder in the process of pneumatic conveying, and calculated and analyzed the parameters of air velocity and seed dynamic inertia in the conveying process, to provide a reference for the design and improvement of the set-row seeder [33].

Smooth seed-guiding technology, also known as zero-speed seed drop, means controlling the seed to achieve zero speed in the horizontal direction as it falls into the seed bed during the descent. A brush-belt seed-guide device developed by John Deere, USA (Figure 4) effectively solves the problem of bouncing seeds falling into the seed bed under high-speed sowing operation conditions [34]. To improve the quality of the high-speed sowing operation, Liu Quanwei designed a toggle-finger synchronous belt-type seed precision delivery mechanism, which realizes the complete control of the seed transport trajectory from the seed metering device to the seed bed, in precision seeding [35]. The USA developed the conveyor-belt seed guide for the speed tube for a no-till seeding operation (Figure 5); each partition takes single seeds from the seed metering device. The speed of the partitioned seed delivery belt is adjusted with the operating speed to achieve zero-speed seed delivery [36].

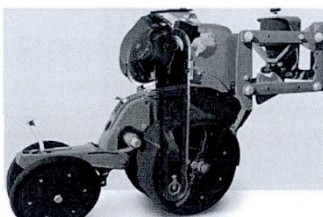

**Figure 4.** Brush-belt type seed-guide device.

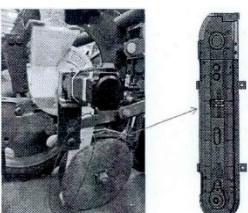

**Figure 5.** Conveyor-belt type seed-guide device.

Under high-speed seeding operations, the seed falling into the seed bed with precision is essential to ensure uniform seeding. Through research on seed-guide tubes for seed transportation, the restraint effect of seed-guide tubes on seeds can reduce the kinetic energy of seeds falling into the seed bed and reduce seed collision in the seed bed.

The falling of the seed into the seed bed using precision technology mainly reduces the bounce of seeds when they fall into the seed bed. Because seeds are discharged from the seed discharging device of the precision seeder and fall into the seed bed through the seed-guide tube, the seeds have specific kinetic energy at this time. The seeds may bounce or roll when falling into the seed bed and touch the soil, resulting in uneven distribution of seed spacing in the seed bed [37]. Seed pressing devices are often used to press the seed falling into seed beds into position, in a timely way. The commonly used seed-pressing devices mainly include the seed-pressing tongue and the seed-pressing wheel, primarily used in pneumatic precision seeders. For example, Wang Yunxia et al. designed a seed-pressing device (Figure 6) applicable to a precision seeder with airflow-assisted high-speed seed delivery, which can effectively reduce the inconsistent grain spacing caused by the collision of seeds in the seed-guide tube [38]. Precision-seeding research in the USA has developed a plastic wear-resistant seed-pressing tongue (Figure 7) that enhances the seed contact with the soil in the seed bed and effectively positions the seed in the seed bed [22].

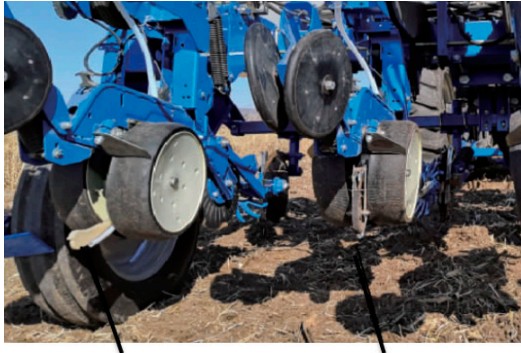

seed-pressing tongue     seed-pressing  wheel

**Figure 6.** Precision seeder seed-pressing device.

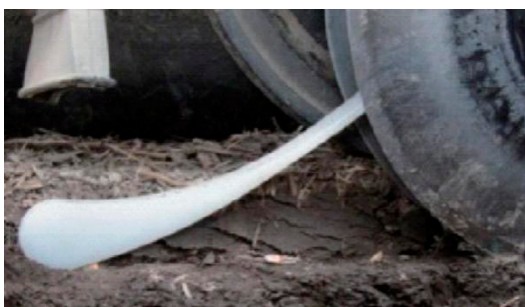

**Figure 7.** Plastic seed-pressing tongue.

Much research has been carried out on the seed metering device, which is reflected in the design of the seed metering device and the improvement in working performance. At the same time, related research has also been carried out on the link between seed guidance and seed dropping. Forming an indispensable link to precision seeding, the research into the regulation technology for a consistent sowing depth is particularly important.

*2.2. Consistent Sowing-Depth Regulation Technology*

Precise seed discharge technology, smooth seed-guiding technology, and the seed falling into the seed bed with precision in the above Precision seeding operation, aim to achieve grain-spacing consistency by regulating the seed migration process [39]. Sowing depth is an essential factor affecting the emergence of crops; in line with the crop, agronomy sowing depth conditions the healthy sprouting, growth and development of harvests; if the sowing depth is too shallow, it is not conducive to the seeds fully absorbing the soil nutrients, the root system is not fully developed, leading to the seedling being weak and prone and a lack of seedlings, with other situations which easily cause yield a reduction in the yield; if sown too deep, it is not easy for the seed sprout to break through the soil surface layer, resulting in the late emergence of weak seedlings, uneven emergence, and other situations [40].

During the precision seeder operation, the device's vibration is caused by the change in soil resistance of the devices in contact with the soil, resulting in a difference in the sowing depth. Therefore, the research into soil covering and compacting under the agronomic requirements to ensure the consistency of sowing depth provides the theoretical basis for designing and modifying the devices in contact with the soil precision seeder and then realizing the purpose of improving the quality of the sowing operation and increasing the profit. It is essential to enrich and enhance the soil covering and compacting technology, improve the reliability of the devices in contact with the soil of the seeding machine, and realize the sowing effect of a consistent sowing depth. Foreign agricultural equipment companies, such as John Deere, developed precision seeders mainly by installing terrain-adapting wheels on both sides of the furrow opener to regulate the depth of the furrow according

to the terrain, to achieve the purpose of a consistent sowing depth. At the same time, some other foreign companies have detected the depth of furrowing through the sensor installed in the seeding unit. Suppose the depth of furrowing does not meet the operation requirements. In that case, the hydraulic system will regulate the planting monomers to achieve real-time regulation of the furrowing depth to achieve consistent sowing depth. The foreign precision seeder achieves consistent sowing depth, mainly through the four-bar construction, combined with the terrain-adapting wheel and sensor-assisted monitoring, ensuring that the accuracy of terrain adaptation can be applied to high-speed operation in large fields. Nevertheless, the foreign precision seeding operational mode, environment, and domestic differences, which make this alien technology for sowing depth control, can only partially apply to the sowing depth control in China. Relevant research has been carried out in China for sowing depth consistency; for example, Zhang Rui designed a sowing depth control mechanism for a corn no-till seeder using microcontroller control, wireless communication, and other technologies to ensure high consistency of sowing depth in the corn seeder, which effectively improved the quality of the sowing operation [8]. Feng Xue designed a sowing depth control device for potato precision seeders in hilly mountainous areas, effectively enhancing the potato sowing depth uniformly; the device operation had good stability [41].

From the above analysis, the research on precision seeding at home and abroad is divided into four areas. The first one is the precision seed discharge, and the current research on the seed metering device; the pneumatic seed metering device has a better operational performance than the mechanical seed metering device, which is more suitable for research into the precision seed metering device. Meanwhile, an electric-drive seed metering device will replace the mechanical-drive precision seeding in the development of precision seeding technology, due to its excellent characteristics of being suitable for a high-speed and highly precise seeding operation [6]. The second is the smooth seed-dropping; the seed-guide tube has a direct impact on the trajectory and speed of the seed falling into the seed bed, although the transformation of the shape of the seed-guide tube cannot overcome the sizeable relative velocity between the seed and the seed bed caused by bouncing or rolling problems. At present, the research into adding a stable migration device to a seed-guide tube can effectively achieve zero-velocity seeding but also increases the structure complexity, so the simplified optimization design of the seed-guide tube is a new research direction. The third is the seed falling into the seed bed with precision; the fall into the seed bed with precision technology research started late, and there is relatively little research; the current setup of the pressure plate or pressure seed wheel can effectively achieve the restraint of the seed, and there is currently more information regarding pneumatic seed metering for starting research. The fourth is consistent sowing depth consistent. The formation of sowing depth is mainly the result of the joint operation of several links, including furrow opening and soil return during covering and compacting. To ensure consistent sowing depth, it is necessary to ensure the terrain-adaptive furrowing, stable soil covering, and suitable real-time compacting.

With the application of GPS, GIS, automatic control, sensor detection, and other technologies in the field of precision seeding, the existing precision seeding technology needs to be continuously developed to meet the needs of modern agricultural industrialization. There is a need to establish a joint tillage-operation and precision seeder, such as precision seeding and tillage, a precision fertilization joint in the operation of machinery for the developmental direction, and sizeable high-speed operation as a development trend to replace the original small and medium-sized precision seeder. At the same time, the application of precision seeding under no-till sowing conditions should be expanded to increase the scope of the application.

## 3. Precision Sowing Conditions of consistent Sowing-Depth Regulation

Precision seeding technology has been developed in developed countries for decades, from the initial precise seed discharge and smooth seed-guiding technology of the seed

rower to the research on the seed falling into the seed bed with precision technology, gradually developing to consistent sowing-depth regulation, and now to the variable sowing, according to the soil environment. At present, China's precision sowing technology research needs to catch up. At this stage, China's precision seeding technology includes specific research into the precise seed discharge, smooth seed-dropping, and the seed falling into the seed bed with precision. More and more attention is paid to the consistency of sowing depth. The formation of the sowing depth results from the joint action of the furrow opener and the soil covering and compacting, so the soil covering and compacting, which consistently have an important influence on the sowing depth, will also become the development direction of our precision seeding technology. The furrow-opener depth control method is mainly realized by the imitation mechanism driving the sowing monomer to undulate with the terrain; at present, the main ways of the sowing monomer regulating the furrow-opener depth are as follows (Table 2):

**Table 2.** Sowing monomer adjustment method.

| Type | Advantage | Disadvantages |
| --- | --- | --- |
| Mechanical type | Simple structure and low cost | Poor adaptability, inconvenient adjustment, and inability to regulate town pressure in real time |
| Pneumatic type | No over-compaction and town-pressure regulation | Slow response time |
| Hydraulic type | Faster speed adjustment of town pressure is possible | More complex structure than mechanical type |

As an essential part of guaranteeing consistent sowing depth, stable soil covering and proper soil compaction will become vital elements of precision-seeding technology research.

### 3.1. Consistent Sowing-Depth Regulation Technology—Soil Covering

The purpose of the soil covering device is to protect the seed that falls into the seed bed, creating an environment for seed germination and growth that directly impacts seed spacing and seedling emergence. According to the current test results, the structural and operating parameters of soil-covering devices directly influence the performance of soil covering. High soil-covering quality is required to ensure consistent thickness and uniform soil-covering volume during precision seeding operations [42]. The ordinary soil-covering device has three classifications: disc, extruded, and plate type soil-covering device (Figure 8).

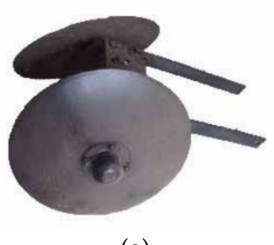 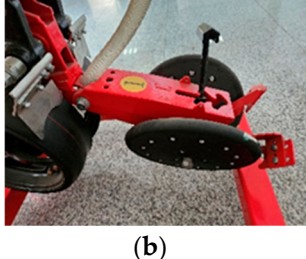 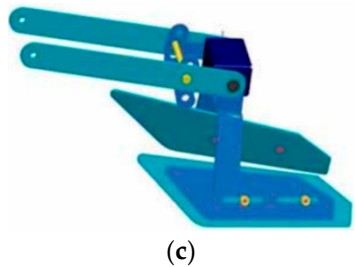

(**a**)        (**b**)        (**c**)

**Figure 8.** Soil-covering device. (**a**) Disc soil-covering device. (**b**) Extruded soil-covering device. (**c**) Plate-type soil-covering device.

Li Yanli designed the disc soil-covering device consisting of two discs arranged at a specific angle. Its structure is adjustable, so the amount of soil covering can be adjusted by changing the device's opening and tilt angles. The spacing between the double discs can be changed according to different operating conditions to meet the actual operating requirements of varying seed furrows [43]. In addition, the device has the advantages of low resistance and stable operation in the process of operation.

The extruded soil-covering device's main structure comprises two opposing angled wheels, which can be adjusted by a tie rod before an operation to change the soil compacting and covering effect. The main factors affecting the operating outcome are the operating speed and the size of the wheel-body clamp angle [44].

The plate-type soil-covering device can be used in a corn seeder; part of the device can be adjusted, according to the shape of the open furrow and the size of the required mulch thickness, and with the characteristics of simple structure, the factors affecting the soil-covering effect are soil-covering inclination and the opening angle [45].

As the part is in contact with the soil of the seed bed arrangement, the soil-covering-device operation can provide the material conditions such as moisture, temperature, and the nutrients required for seed germination, which profoundly impacts the yield and quality of crops. Its soil covering technology and the performance of its equipment directly affect the quality of seeding and operational efficiency. Scholars and experts in China and abroad have researched the problems of the soil cover's poor mass and stability problems. For example, Song Lupeng et al. modeled the cutting soil dynamics of the disc soil-covering device and analyzed the relationship between cutting stress and cutting angle and shear rupture angle, during the operation of the disc soil-covering device and based on the SPH meshless method. The simulation of the optimal parameter ratio effectively analyzed the displacement of the seeds after touching the soil, with the change in soil flow movement during the operation of the disc soil-covering device [46]. Liu Xuanwei et al. used a uniform design method to design a double-layer disc-type soil-covering device to reduce the displacement of seeds during soil covering, and conducted tests and evaluations on the prepared soil-covering device [47]. Yao Lintao et al. designed a fertilizer soil-covering device suitable for hilly orchards, which can realize the fertilizer-deep application and complete soil-covering operation simultaneously, through the structural combination of an externally grooved wheel-type fertilizer discharger and spiral churning soil-covering device [48]. Sun Wei et al. designed a film soil-covering device using a scraper lift-belt type. By actively transporting the soil for soil covering, the operational efficiency and soil-covering quality were improved [49]. Chao Baoba et al. designed a device for automatically adjusting the soil covering the amount of the transplanting machine. The device obtained the information on the soil-covering amount on the film by measuring the soil thickness while working and controlling the depth of the soil-covering disc under the treatment of the control module to adjust the soil-covering amount and reduce the inconsistency of the soil-covering thickness caused by the uneven ground of the soil-covering device [50]. The current foreign research in soil covering focuses on extruded soil-covering devices, which have a soil-covering and compacting function, and is research based on the same principles in different structural forms.

Soil is the primary environment for nutrient supply for crop growth; studies were conducted to investigate the effects of different covering materials on seed germination and development under the same covering conditions. For example, to examine the impact of other covering materials on the yield of large bulbous cap mushrooms, Zhan Meirong et al. used peat soil, field soil, peat soil + field soil, volume ($m^3$) ratio 1:2 composite soil, and grain hull + field soil, volume ($m^3$) ratio 1:2 blended soil as covering materials for Stropharia rugoso-annulata. The test results showed that the Stropharia rugoso-annulata emergence was the fastest. The yield and economic benefits were the highest under the soil-covering conditions of grain hull + field soil composite soil [51]. At the same time, according to the theme of "soil covering" in the China National Knowledge Infrastructure, 13,400 articles were searched, and the first three studies focused on cultivation techniques such as the soil covering of mushrooms; the research on mulching under precision seeding is still not a hot spot (Figure 9).

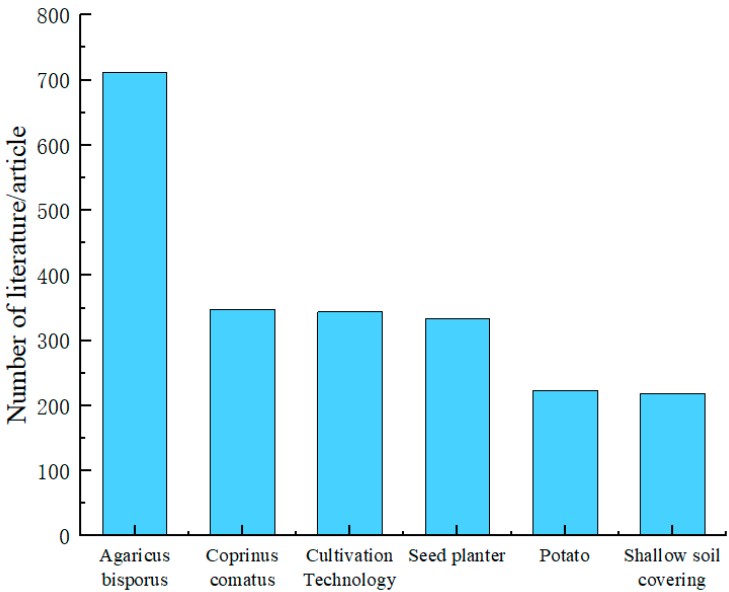

**Figure 9.** The proportion of soil cover in precision-seeding studies.

In summary, the research into soil-covering devices at home and abroad can be divided into three main areas. The first one is the traditional structure optimization design, which generally focuses on the structure of the existing soil-covering device to make micro-innovations. The soil-covering operation under sowing conditions mainly relies on passive methods such as squeezing and gathering to achieve soil coverage of seeds, which is still based on the traditional passive soil-covering operation. The second type is the mechanical type of initial soil covering, designed with a spiral churning soil-covering device to effectively realize the initial nature of the soil-covering process. For complete film soil covering, it is widely used for crops such as sesamum and potatoes, which have the advantages of drought resistance. However, there are problems of uneven soil covering and misalignment of cavity holes caused by low soil-covering volume in the way of a simple machine-laying film and manual soil-covering operation. The third is the soil-covering-volume regulation and control, soil-covering-volume automatic adjustment device, and the sowing operational process of quantitative soil covering to achieve consistent sowing depth, which is specifically referenced. The second and third kinds have produced little research on precision seeding operations, mainly focusing on film soil covering and orchard-fertilization soil covering. At present, the precision seeding operation still relies on the traditional soil-covering device operational effect, with the lack of soil-covering initiative and soil-covering volume control resulting in unstable soil covering and other situations; this results in weak seedlings, uneven planting and even yield reduction and other problems, seriously restricting the development of sowing technology and its equipment and sowing-quality improvement.

### 3.2. Consistent Sowing-Depth Regulation Technology—Soil Compacting

As the last and most critical part of precision seeding, the compacting wheel compacts the soil after soil covering, and aims to reduce evaporation to avoid frost damage and effectively promote the germination of seeds. The effect of compaction has a direct impact on the quality of sowing. As China is a vast country with significant environmental differences in different regions, the compaction requirements of crops are guaranteed to be met under the premise of meeting local agronomic needs. Soil-compacting devices developed in recent years form a full range of types, with many types of structural forms. The main component of the soil-compacting device is the soil-compacting roller; the current soil-compacting roller mainly exists in the form of cylindrical soil-compacting wheel [52],

concave or convex soil-compacting wheel [53], rubber soil-compacting wheel [54], and conical compound soil-compacting wheel [55] (Figure 10).

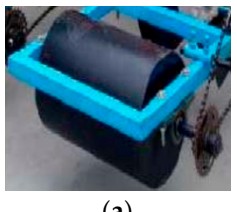 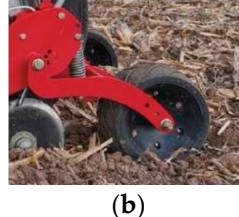 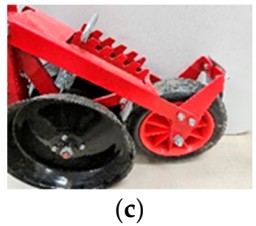 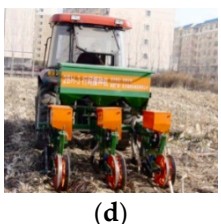

(**a**)  (**b**)  (**c**)  (**d**)

**Figure 10.** Soil-compacting device. (**a**) Cylindrical soil-compacting wheel. (**b**) Concave or convex soil-compacting wheel. (**c**) Rubber soil-compacting wheel. (**d**) Conical compound soil-compacting wheel.

Currently available soil-compacting devices vary in function and structural approach [56], and corresponding studies on the structural design of the soil-compacting wheel are being carried out. For example, Wang Hanyang designed a compact, three-way adjustable soil-covering and compacting device for the practical requirements of a no-till soil-covering precision-seeder soil-compacting device for wheat stubble fields. During the sowing operation, the soil covering the width and soil compacting intensity can be adjusted by the adjustment handle, according to the tillage environment and agronomic requirements [57]. Wang Wenjun designed a soil-compacting device with a function based on profiling function. The surface of the soil-compacting roller is designed with sticky reduction and an anti-slip structure, with the pressure seeding belt for pressure profiling adjustment. At the same time, this reduces the phenomenon of slipping of the soil-compacting wheel during the operation [58]. Zhao Jiale designed a flexible soil covering and compacting wheel, which can perform the profiling operation of the precision seeder and can realize the uniform distribution of soil-compacting pressure in different terrains [59]. However, in the actual operational process, the soil-compacting roller slippage of the soil-compacting device has become the primary problem to be solved by scientific researchers. Two main methods exist to solve the slippage problem: traditional and bionic viscosity reduction and resistance reduction. The following table compares the traditional viscosity reduction and drag reduction and the bionic viscosity reduction and resistance reduction methods (Table 3):

**Table 3.** Comparison of traditional viscosity reduction and resistance reduction and the bionic viscosity reduction and resistance reduction methods.

| Methods | Type | Classification | Principle of Operation |
|---|---|---|---|
| Traditional viscosity-reduction and resistance-reduction. | Mechanical type | Scraping method | The scraping part on the soil-touching part removes the adhering soil. |
| | | Vibration method | By applying vibration to the part that adheres to the soil, the soil is continuously shaken and dislodged from the part. |
| | Surface modification | Whole or partial-change method | Overall shape change or local area geometrical change on the overall surface of the earth-touching component. |
| Bionic-viscosity-reduction and resistance-reduction. | Structural Bionics | Biological non-smooth bionic | Structural design of devices that mimic the unique body shape and form of living things. |
| | | Biomorphic Biomimicry | Body shape (shape, form) biomimicry, conformation that mimics the shape of a living organism. |
| | Functional bionics | Bio-flexible bionic | Mimicking the non-linear deformation of living organisms, it can recover its original position and realize the function of decoupling and reducing resistance through bio-flexible recovery. |
| | | Elasticity method | Change in the degree of deformation of the contact surface, so that the soil falls off during the deformation of the component. |

As the application of the bionic method in agricultural machinery becomes more widespread, more bionic soil-compacting devices with excellent performance are emerging, such as that designed by Chang Yuan et al., which is a soil-compacting roller with a bionic convex package geometry, using the head of the dung beetle as a bionic prototype. This effectively realized the reduction in viscosity and resistance in the soil-compacting process by bionic biomorphic body shape, with the attendant advantages [60]. Faced with the problems of slippage and adhesion of the soil-compacting wheel and increased resistance caused by the actual field operation, Jia Honglei et al. designed the bionic elastic soil-compacting roller by borrowing the flexible geometric features of the earthworm surface as the prototype. It has many advantages in viscosity reduction and resistance reduction, such as the uniform distribution of compacting pressure and a low slippage rate [61].

At present, the soil-compacting device in the traditional viscosity-reduction and resistance-reduction technology is mainly seen in the soil-compacting roller research, based on the existing device to optimize the design [62], while the bionic-viscosity reduction and resistance-reduction technology is designing a new device with the principle of the structure and function of living organisms [63]. In terms of soil-compacting pressure regulation, the real-time regulation of the soil-compacting pressure of the soil-compacting roller is in the initial stage, and most soil-compacting devices do not have the real-time adjustment function. A few realize the real-time adjustment function, but the accuracy needs further improvement.

Jia Honglei et al. proposed a real-time measurement method based on soil firmness for the problem of the complex and inaccurate measurement of the compacting pressure of the soil-compacting device in the seeding operation. The model can achieve accurate real-time measurement of the soil-compacting pressure, which provides a technical basis for the subsequent research on real-time regulation of the soil-compacting pressure. The detection method did not achieve the regulation of the soil-compacting pressure of the ballast roller, and only reached the real-time extraction of the soil-compacting pressure [64]. Bai Huijuan conducted a study on the soil-covering and compacting device of a corn seeder to achieve indirect control of the sowing depth during operation by real-time adjustment of the hydraulic pressure on the profiling mechanism and real-time regulation of the soil-compacting pressure by adjusting the spring elongation of the device's soil-compacting mechanism to ensure the requirement of soil compactness during operation [65].

Compared to the earlier domestic research, foreign precision-seeding monomer development, especially the soil-compacting device, can achieve a soil compacting operation in high-speed operating conditions. The research is becoming more and more mature. The existing research on soil-compacting wheels in China is based on the low- and medium-speed operating environment; promising research results have been achieved in recent years. However, in the high-speed operating environment, the research on domestic soi- compacting devices has a large gap in comparison to foreign countries. The John Deere Company designed a rapid adjustment device that can change the relative angle of the soil-compacting wheel and the soil-compacting pressure [66]. RFM AG developed a spring-coil soil-compacting device that can automatically de-soil and perform soil sealing during operation [67]. Schaffert designed a claw-type soil-compacting device that does not adhere to excessive soil, and can be used for various large seeding units. The operation can be achieved by compacting the seed and soil in the center so that they are in close contact. Moreover, the side column can break the soil surface and avoid surface over-compaction [68].

In summary, the research on soil compacting devices at home and abroad can be divided into three areas. The first one is the design of the soil-compacting wheel to reduce viscosity and resistance. The overall innovative research is carried out by bionics, using insects and animals as reference prototypes to design the soil-compacting wheel with reduced resistance, to avoid slippage [69]. The second is the soi-compacting pressure regulation, the design of the device to effectively realize the sowing operation process of real-time control of the sowing depth and compaction, but only if the device has both soil-

covering and compaction functions and there is room for wireless transmission upgrades in the system control. The third type is soil compacting under high-speed operating conditions; the research is mainly carried out in foreign countries first; the soil-compacting device can realize the soil-compacting operation under high-speed operating conditions, and the investigation is becoming more and more stable. However, the soi- compacting pressure dispersion reduces in the soil during the downward pressure of the soil-compacting wheel, resulting in a situation where the top is solid, and the bottom is loose. There needs to be more research on this situation. At the same time, with the popularization of high-speed seeding operations abroad, the existing soil-compacting methods in China can hardly meet the needs of the current development in the face of high-speed operations.

### *3.3. Future Development Issues and Dynamics*

3.3.1. Soil Covering and Compacting Technology Developmental Issues

A soil-covering and compacting operation directly affects the inconsistency of sowing depth during the precision seeding operation. Due to the complex operating environment and uneven surface height [70], the current soil-covering device needs to effectively realize the adjustment of the soil-covering amount. The soil-covering operation is mainly realized by squeezing and convergent reflux. Most of the research focuses on the soil-compacting operation, but there needs to be research into the quantitative soil covering. The actual effect on the sowing depth is mainly the combined operational effect of soil covering and compacting, so it is essential to realize active quantitative soil covering to control the sowing depth. According to the current research on soil covering and compacting, the following problems are mainly concentrated upon:

(1) Precision-seeder soil-covering and compacting components for different operating environments lack the corresponding adaptability to soil covering, and basically, all companies are a product applied to various scenarios; at the same time, soil-compacting devices on different soils also have interoperability, which affects the lack of data accumulation and research.

(2) In the soil-covering operation, the soil covering is mainly achieved by convergent reflux and other rough forms of soil covering, which is prone to uneven soil covering in the complex field environment, and which brings specific difficulties to the subsequent soil-compacting operation.

(3) The contact soil components of domestic enterprises are mostly imported imitations, lacking core competitiveness in material technology. At the same time, there is a shorter service life than abroad, and the parts are not strong enough to cause damage in the complex field operating environment, easily affecting the operation, and so on.

(4) At present, there is a lack of research into soil-covering devices, while the real-time regulation of soil compacting is in its initial stage. The development of intelligent soil-contact components for sowing depth control is a weak point.

(5) With the popularization of high-speed precision-seeding technology, the technical problem of the corresponding domestic soil-covering and compacting devices effectively matching the performance requirements under high-speed working conditions has yet to be solved.

3.3.2. Development Prospects of Soil-Covering Active Compacting

In summary, the development of precision-seeding technology, the control of the contact parts of the soil covering, and compacting for sowing depth control plays an important role in ensuring and improving the quality of the operation. As agricultural technology advances and research into soil and contact components continues, more functional, efficient, and cost-effective soil-covering and compacting technologies are being developed and enriched. Researchers are currently conducting research into soil-component interaction analysis, structural design optimization, material selection, and intelligent control of soil-compacting components in order to reduce operating costs, the working resistance of soil-compacting components and improve operating efficiency.

At the same time, the soil-covering and compacting devices as a device in contact with the soil need to meet the development needs of modern agricultural mechanization. In addition, in China's different regions, different crop types and agronomic requirements, and other conditions of the actual needs of the production situation, together with China's soil-covering and compacting devices in the design of the contact parts require optimization and the promotion of a certain degree of difficulty and challenge. In order to adapt to the developmental needs of China's seeding operations, the development of soil-covering and compacting technology has taken the following directions [71].

(1) Given the existence of soil characteristics, cropping patterns, crop varieties, and other existing differences in different regional geographical environments in China, the analysis clarifies the interactions between different soil-crop-touch components and develops regionally adapted soil-covering and compacting devices [12].

(2) Improving enterprises' processing and equipment technology through research in the material science of soil-covering and compacting touchdown components. To ensure the operation effect, the strength and life of the soil-covering and compacting contact parts should be improved.

(3) Strengthening the optimization design of the structure of the soil-covering and compacting device through the soil-covering and compacting device in the process of operation to achieve viscosity reduction and resistance reduction, to ensure the operating effectiveness of the premise of effectively reducing power consumption and loss of soil contact parts.

(4) Based on the traditional structure design [12], the research into the control method of soil covering and compacting is carried out by combining mechanical design and electrical control. Promoting the intellectual development of agricultural equipment and improving the operational performance and stability of the device.

(5) Relying on the developmental trend of the industry, research into real-time regulation and control of a quantitative soil-covering active compacting link, through the close combination of agricultural machinery and agronomy, under the conditions of high-speed precision-seeding operation to further improve the operational quality and efficiency.

## 4. Conclusions

With the introduction and promotion of the precision agriculture concept, agricultural production is developing towards a standardized scale and way of operation [72]. Precision seeding, as one of the ways to realize precision agriculture, is essentially the improvement of mechanized planting quality, and has significant development prospects in different tillage methods, mainly through including precision seeding in various modes such as the no-till operation, compound operation, and high-speed operation; at the same time, combined with relevant frontier technologies such as prescription map, sensor detection, GPS positioning, and unmanned driving, precision seeding has diversified developmental directions. Although the precision seeding conditions for soil-covering and compacting technology have been developed and perfected, there are still some problems to be solved to meet the evolving technological needs and sowing quality requirements.

(1) The soil-covering operational device of a large amount of soil covering will lead to excessive soil flow velocity on the seed bed impact, meaning that the seed displacement and grain-spacing consistency are affected; at the same time, the amount of uneven soil covering too much mulch will cause the dry soil and wet soil to be mixed, so that the seed cannot not germinate in the damp soil environment. In the actual operation process, the soil-covering process is carried out once. No secondary adjustment can be made, so achieving a stable and consistent soil-covering depth in the one-time soil-covering process is necessary.

(2) The high-speed operation of precision seeding in China also provides an excellent test for the soil-compacting effect of the soil-compacting device. As the operating speed increases, the soil-compacting time of the unit area of the soil-compacting

wheel decreases. The lack of soil-compacting pressure means that the seed and soil contact is not close enough, affecting the seed germination. At the same time, the soil-compacting pressure is constantly decreasing in the soil, and there needs to be more research on how to achieve a suitable and consistent soil-compacting pressure around the seed bed and on the soil surface.

**Author Contributions:** Conceptualization, W.Y. and J.H.; methodology, W.Y.; software, C.L.; valida-tion, C.L., H.L. (Han Lin) and H.Y.; formal analysis, H.L. (Han Lin); investigation, W.Y.; re-sources, C.L.; data curation, H.Y.; writing—original draft preparation, W.Y.; writing—review and editing, H.L. (Han Lin), H.Y. and H.L. (Hang Li); visu-alization, C.L.; supervision, J.H.; project administra-tion, J.H.; funding acquisition, J.H. All authors have read and agreed to the published version of the  manuscript.

**Funding:** This research was funded by the Strategic Priority Research Program of the Chinese Academy of Sciences (XDA28010402).

**Institutional Review Board Statement:** For studies not involving hu-mans or animals.

**Informed Consent Statement:** Not applicable.

**Data Availability Statement:** The data in Figure 9 are from China National Knowledge Infrastructure.

**Conflicts of Interest:** The authors declare no conflict of interest.

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
