# Peer review of "Current Situation and Future Development Direction of Soil Covering and Compacting Technology under Precision Seeding Conditions in China"

_applsci, doi:10.3390/app13116586_

Round 1

Reviewer 1 Report

This article reviews existing high-speed corn and soybean metering devices, zero-speed seed drop technology, soil covering devices, and soil compacting wheels.  The report is well organized and the existing problems of each part of the seeders are well discussed. However, the introduction is hard to understand. Language editing is required.

- There is a problem demonstrating figures. It would be better to mark figures as a, b, c, and d and describe them rather than giving numbers to each figure all over the manuscript. 

Pneumatic-suction type working process was described, what about others?

-Please choose the adequate name for the device in Figure 17. 

Language editing is required!

Reviewer 2 Report

1. Authors through this paper has explained the critical role of Precise Seeding technology in sowing quality improvement .The promotion and application of Precise Seeding technology are also pointed out as an  essential means to achieve abundant crop production and harvest.

2. The topic is considered to be original and quite relevant in the field and it does address a specific gap in the field.

3. This paper adds value to the subject area through the research which clearly depicts the Precise seed discharge technology, Smooth Seed Guiding Technology, and Fall into the seed bed with precision and Sowing depth consistent regulation technology to ensure Precise Seeding 

4. Methodology carried out is considered to be good as the results drawn are quite satisfying .Improvements can be done only in the elaboration process of all described methodology with few more data and a comparative table can be given for all methodologies adopted. 

5. Conclusion needs to be rewritten against the evidence and arguments which are presented as they are not found to be consistent with the research carried out.

  1.  

Need to carry out few grammatical corrections.

Reviewer 3 Report

The article provides an overview of the methods of sowing and soil compaction. However, several elements are missing:

1. The abstract does not clearly define the purpose of the article.

2. Lack of comparison of the indicated sowing technologies. The most readable form would be a table

3. The same remark as in point 2, only with regard to the technology of soil cover and compaction.

Round 2

Reviewer 1 Report

Please, revise your manuscript according to the journal template!

Language editing is required